# Prognostic accuracy of MALDI-TOF mass spectrometric analysis of plasma in COVID-19

Lucas Cardoso Lazari[1], Fabio De Rose Ghilardi[2], Livia Rosa-Fernandes[1], Diego M Assis[3], José Carlos Nicolau[4], Veronica Feijoli Santiago[1], Talia Falcão Dalçóquio[4], Claudia B Angeli[1], Adriadne Justi Bertolin[4], Claudio RF Marinho[1], Carsten Wrenger[1], Edison Luiz Durigon[5], Rinaldo Focaccia Siciliano[4], Giuseppe Palmisano[1]

**SARS-CoV-2 infection poses a global health crisis. In parallel with the ongoing world effort to identify therapeutic solutions, there is a critical need for improvement in the prognosis of COVID-19. Here, we report plasma proteome fingerprinting that predict high (hospitalized) and low-risk (outpatients) cases of COVID-19 identified by a platform that combines machine learning with matrix-assisted laser desorption ionization mass spectrometry analysis. Sample preparation, MS, and data analysis parameters were optimized to achieve an overall accuracy of 92%, sensitivity of 93%, and specificity of 92% in dataset without feature selection. We identified two distinct regions in the MALDI-TOF profile belonging to the same proteoforms. A combination of SDS–PAGE and quantitative bottom-up proteomic analysis allowed the identification of intact and truncated forms of serum amyloid A-1 and A-2 proteins, both already described as biomarkers for viral infections in the acute phase. Unbiased discrimination of high- and low-risk COVID-19 patients using a technology that is currently in clinical use may have a prompt application in the noninvasive prognosis of COVID-19. Further validation will consolidate its clinical utility.**

## Introduction

The pandemic of SARS-CoV-2 infection, the etiological agent of coronavirus disease 2019 (COVID-19), has affected millions of people worldwide. The first case was reported in Wuhan, China, and as for 30, September, 33,722,075 people have been infected and 1,009,270 died. The ongoing outbreak is considered a pandemic (World Health Organization). The symptoms range from mild with fever, dry cough, headache, fatigue, and loss of taste and smell to severe complications, including difficulty breathing or shortness of breath, chest pain, and loss of speech or movement that can lead to hospitalization and death (1). Although vaccines and small molecule treatments are in clinical trial, no definitive treatment for COVID-19 is available yet (2, 3, 4). A mortality rate of ~4% has been detected in COVID-19 patients compared with 0.1% in influenza infection (World Health Organization). Because of that, it is imperative to identify patients at high risk for severe illness to assist them with supportive therapy. Markers of COVID-19 severity have been proposed (5, 6, 7).

MALDI-MS has been successfully implemented into the microbiology field building reference spectral libraries for rapid, sensitive, and specific identification of bacterial and fungal species (8). This approach is well established and accepted in many countries for routine diagnostics of yeast and bacterial infections. Viral species identification has been elucidated using similar strategies (9). Recently, MALDI-TOF mass spectrometry (MS) analysis of nasal swabs allowed sensitive and specific detection of SARS-CoV-2 infection (10). Moreover, MALDI-TOF MS analysis of human biofluids have been proposed as diagnostic and prognostic techniques in several diseases ranging from cancer, cardiovascular, neurological, and infectious diseases (11, 12, 13, 14, 15).

Although it was previously described, the use of MALDI-MS and machine learning analyses in COVID-19 nasal swabs samples (16), our study shows a new approach for the identification of a plasma proteomic signature obtained from high- (hospitalized) versus low- (outpatients) risk patients with COVID-19 using an accurate, easy-to-perform, rapid, and widespread technology such as MALDI-TOF MS, which is present in several clinical laboratories worldwide. A training and validation dataset allowed the prioritization of discriminant features identified using bottom-up quantitative proteomics. SAA1 and SAA2 proteoforms were differentially expressed between the two groups allowing the implementation of point-of-care diagnostics. More studies, including larger inter-institutional cohorts, are needed to move this marker into the clinic.

[1]Department of Parasitology, Institute of Biomedical Sciences, University of São Paulo, São Paulo, Brazil   [2]Instituto de Medicina Tropical, University of São Paulo, São Paulo, Brazil   [3]Bruker do Brasil, Atibaia, São Paulo, Brazil   [4]Heart Institute (InCor), University of São Paulo Medical School, São Paulo, Brazil   [5]Department of Microbiology, Institute of Biomedical Sciences, University of São Paulo, São Paulo, Brazil

Correspondence: palmisano.gp@gmail.com; palmisano.gp@usp.br

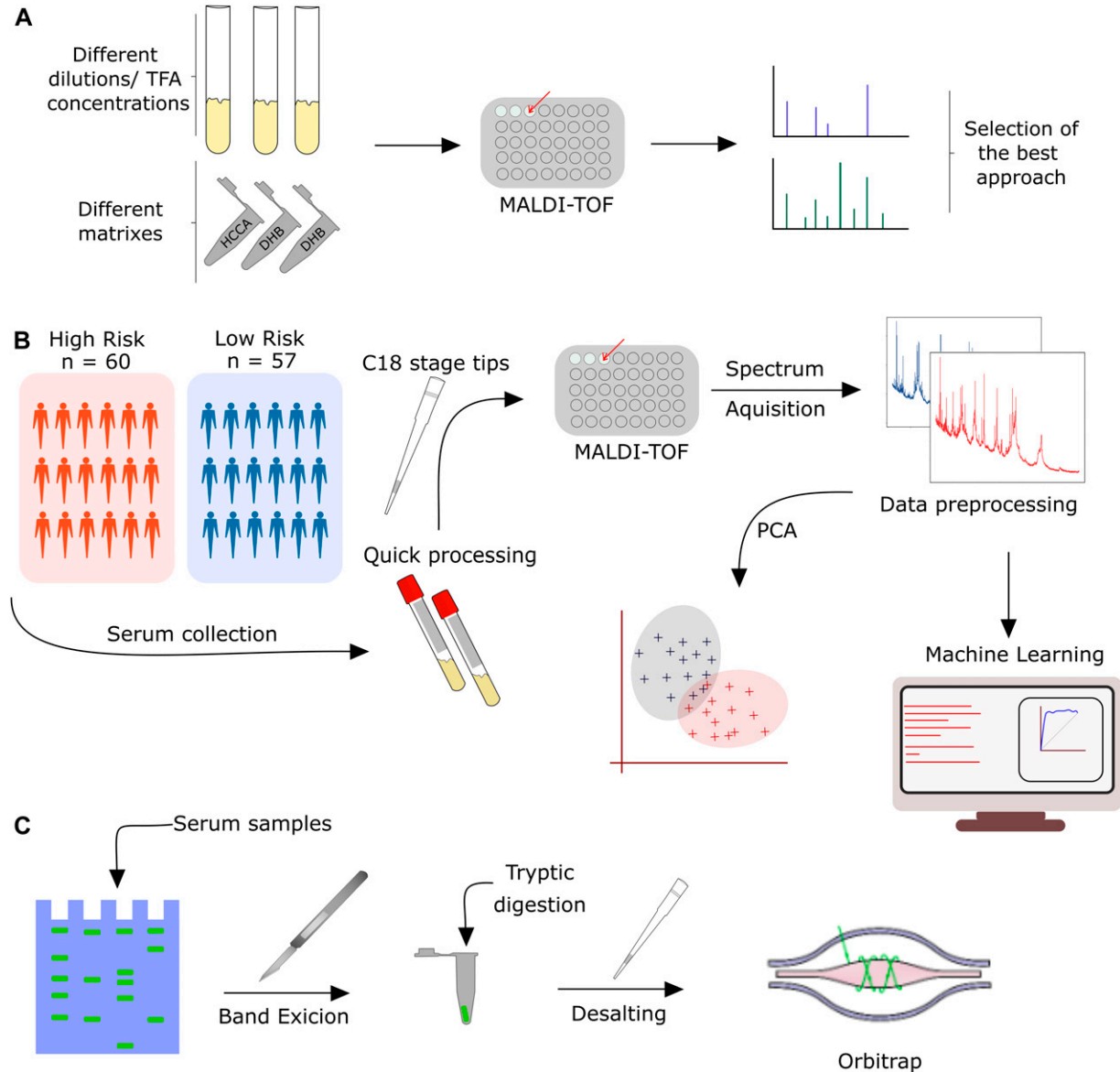

**Figure 1. Experimental workflow applied to this study.**
**(A)** Method development for MALDI-TOF MS analysis of plasma samples. **(B)** MALDI-TOF MS analysis of 117 COVID-19 patients combined with machine learning to identify MS discriminant features in the training and test dataset. **(C)** Biomarker discovery based on 1D SDS–PAGE and nLC-MS/MS analysis.

# Results

### Method optimization and evaluation of reproducibility and variability

The analytical platform shown in this study was developed through three phases: (1) MALDI-TOF MS-based method development for plasma samples, (2) development of a potential clinical application to plasma isolated from COVID-19 patients with high and low risk, and (3) identification of markers to discriminate high and low-risk patients, according to the experimental workflow (Fig 1A–C).

Initial method development focused on the selection of the appropriate matrix using unfractionated plasma. The dried droplet sample preparation method using unfractionated plasma and three matrices (HCCA, DHB, and SA) was tested acquiring the protein/peptide profile in automatic mode. Using the HCCA matrix resulted in the detection of more peaks compared with other matrices (Fig S1). The highest peaks at "m/z" 16,616.3, 13,315.1, 11,095.6, 9,496.2, and 8,316.8 corresponded to human serum albumin with 4–8 charges. To improve the number of peaks detected, the acid concentration within the matrix (TFA 2.5%) was increased (Fig S2). The peak intensity increased and two peaks in the "m/z" 6,000 region were detected; however, the serum albumin peaks were still within the most abundant. To improve the number of peaks detected for each spectrum, C18-based plasma fractionation was performed. The MALDI-TOF performances were evaluated measuring

the number of peaks and the variation in intensity and frequency of specific peaks detected for each spectrum after processing as described in the Materials and Methods section. A total of five plasma samples were tested. C18-based fractionation showed a higher number of peaks detected from "m/z" 2,000–10,000 than unfractionated plasma (Fig S3). A comparison between three matrices showed that HCCA yielded a higher number of peaks and intensity than other matrices (Figs S4 and S5).

Because of that, we chose to perform all analyses using a C18-fractionated sample eluting the proteins/peptides from the micro-column using the HCCA matrix containing 50% acetonitrile and 2.5% TFA as described in the Materials and Methods section. To test if sample pretreatment induces potential artifacts that affect the reproducibility of the entire strategy, quadruplicate analysis of the same spot and analysis of each sample on three different preparations was performed (Fig S6A–C). Two "m/z" regions were selected to calculate the coefficient of variance (CV) considering the sample preparation and MS acquisition variability. Within the "m/z" 5,700–5,900 region, an average of 10% CV was obtained. Within the "m/z" 11,300–11,700 region, an average of 20% CV was obtained (Fig S7). The lower CV in the higher "m/z" region is associated with the lower intensity, which increases the variance within and between samples. The CVs data obtained are in agreement with other reports using MALDI-TOF MS profiling of biofluids (16, 17, 18). Because of that, the optimized sample preparation strategy applied to this study was based on 1 µl of plasma fractionated using C18 micro-columns and proteins/peptides were eluted using HCCA matrix containing 2.5% TFA.

## Prognostic value of the plasma proteome profile of COVID-19 patients

In the next phase, we applied the optimized strategy to plasma samples collected from COVID-19 patients. 117 patients with laboratory-confirmed COVID-19 disease were enrolled in this study, 57 with mild disease, who did not need hospitalization (low risk), and 60 being admitted in the hospital (high risk). Their status was assessed using a combination of molecular, serological, and clinical examination. RT-PCR and ELISA were used to test the active or past SARS-CoV-2 infection. Table 1 shows the demographic characteristics of these two groups of patients. The median age was significantly higher in the hospitalized group (52 yr; IQR 39.5–64.5) than the mild group (35 yr; IQR 29–47). A total of 40 (70%) of outpatients were female and only 28 (47%) were female in the hospitalized group. The median time of symptom onset before blood sampling was also higher in the severe group (9 d and IQR 7–14) than in the outpatients group (4.5 d and IQR 3–6.5 using two-tailed Mann–Whitney U-test). Four patients (3.5%) died (Fig S8A and B). The sex and age distribution observed in this study is in line with the literature findings. Indeed, a significant association between sex, age, and COVID-19 disease prognosis has been reported (19). Male patients have a higher mortality rate, hospitalizations, and lower chance of recovery compared with females (19, 20, 21). It has been shown that female patients have higher plasma levels of IL-8 and IL-18 cytokines and different immune cells number and type sustained along the life that reduce the severity of COVID-19 (22). The most prevalent symptoms in this cohort were fever and myalgia, in patients that were hospitalized with dyspnea (77%) and cough (68%) were the main clinical features. In the group of outpatients, upper respiratory signs as rhinorrhea (82%) and headache (81%) were more prevalent (Table 2). The difference between groups was statistically significant (chi-square test $P <$ 0,005) regarding the type of symptom presented: as rhinorrhea (82% of the mild symptomatic patients and 22% of admitted in hospital patients), headache (81% of the mild symptomatic and 45% of hospitalized patients), and myalgia (88% of the mild symptomatic and 58% of hospitalized patients) being more prevalent in patients not requiring hospitalization. On the other hand, dyspnea was present in only 22% of the mild symptomatic patient and in 77% of the patients that were admitted in a hospital ($P < 0.005$). Patients with a mild presentation of COVID-19 (low risk) presented running nose, headache, and myalgia and did not develop the inflammatory syndrome. This group of patients were more attentive to their first symptoms and search for health care earlier (most of them are health-care professionals which may represent a bias). The most prevalent comorbidities found in our patients were obesity (4% of the mild group and 27% of the hospitalized group) and dyslipidemia (5% of the mild group and 17% of the hospitalized group). Although there was no difference in dyslipidemia's prevalence between both groups, we found a higher number of obese patients (body mass index > 30) between people requiring hospitalization (16/60 = 27%), Table S1. It is important to mention that this survey was carried out at INCOR hospital, an institution specialized in heart and lung diseases, so we did find a higher prevalence of cardiovascular diseases than in the general population (as heart transplantation patients and people with chronic conditions).

**Table 1. Epidemiological characteristics of the 117 COVID-19 patients investigated in this study.**

| Individuals, no. | Low risk (N = 57) | High risk (N = 60) |
|---|---|---|
| Sex, no. (%) | | |
| Male | 17 (30) | 32 (53) |
| Female | 40 (70) | 28 (47) |
| Age (yr), mean (SD) | 37.5 (11.2) | 51.8 (16.5) |
| Median (yr) | 35 | 52 |
| Time onset symptoms (days) until sampling, median (IQR) | 4.5 (3–6.5) | 9 (7–14) |

SD, standard deviation; IQR, interquartile range.

**Table 2. Clinical findings associated with the 117 COVID-19 patients investigated in this study.**

| Symptoms | Low risk | High risk | |
| --- | --- | --- | --- |
| | (N = 57) | (N = 60) | |
| Fever, no (%) | | | |
| Y | 39 (68) | 47 (78) | P = 0.2 |
| N | 18 (32) | 13 (22) | |
| Headache, no (%) | | | |
| Y | 46 (81) | 27 (45) | P < 0.005 |
| N | 11 (19) | 33 (55) | |
| Diarrhea, no (%) | | | |
| Y | 14 (25) | 24 (40) | P = 0.075 |
| N | 43 (75) | 36 (60) | |
| Myalgia, no (%) | | | |
| Y | 50 (88) | 35 (58) | P < 0.005 |
| N | 7 (12) | 25 (42) | |
| Dysgeusia, no (%) | | | |
| Y | 27 (47) | 31 (52) | P = 0.642 |
| N | 30 (53) | 29 (48) | |
| Anosmia, no (%) | | | |
| Y | 20 (35) | 21 (35) | P = 0.992 |
| N | 37 (65) | 39 (65) | |
| Running nose, no (%) | | | |
| Y | 47 (82) | 13 (22) | P < 0.005 |
| N | 10 (18) | 47 (78) | |
| Dyspnea, no (%) | | | |
| Y | 22 (39) | 46 (77) | P < 0.005 |
| N | 35 (61) | 14 (23) | |
| Expectoration, no (%) | | | |
| Y | 10 (18) | 14 (23) | P = 0.43 |
| N | 47 (82) | 46 (77) | |
| Cough, no (%) | | | |
| Y | 40 (70) | 41 (68) | P = 0.829 |
| N | 17 (30) | 19 (32) | |
| Fever, no (%) | | | |
| Y | 39 (68) | 47 (78) | P = 0.225 |
| N | 18 (32) | 13 (22) | |

Chi-squared statistical analysis for each parameter was performed and the *P*-values reported.

MALDI-TOF spectra obtained from the C18-fractionated plasma from the high- and low-risk groups were analyzed, as described in the Materials and Methods section. Analyzing 117 plasma samples, the data preprocessing (peak picking and filtering) yielded 65 peaks detected, which dropped to 49 after the normality and Wilcoxon rank sum test corrected using the Benjamini–Hochberg method for multiple hypotheses testing with an adjusted *P*-value < 0.05. Permutation analysis was performed and returned a false discovery rate < 0.05, demonstrating that the observed differences are not by chance. After Ig filtering of whole dataset for PCA, 38 peaks were identified (Table S2). PCA analysis of significant peaks and Ig-filtered peaks are presented in Fig 2A and B. The Ig filtering demonstrated a slightly better separation than the PCA with all significant peaks.

Ig filtering for the machine learning resulted in 33 peaks detected in fold "1," 35 in folds "2" and "3," and 34 peaks in fold "4" (Table S2). The MS peaks obtained with and without Ig filtering were analyzed using six machine learning algorithms to discriminate between the two conditions under optimized parameters (Table S3). In general, all models did not differ significantly from each other and presented a robust behavior comparing each fold. The results of the six tested models in the dataset without Ig filtering showed that the random forest (RF) had the higher mean area under the curve (AUC) for the ROC curve (0.97) (Fig 3A); in addition, support vector machine polynomial (SVM-P), naïve Bayes (NB), and RF had the lowest SD for AUC values between each fold (Tables S4 and S5). However, the RF model had a better accuracy, sensitivity, and specificity, thus considered to have the best performance of the six tested. For Ig filtering approach, RF had the highest mean AUC for ROC (0.97) with a low SD between each fold (Fig 3B); The RF model had higher accuracy and sensitivity. Because higher sensitivity is more desirable for prognosis, RF model was considered the model with the best performance for Ig-filtered peaks. Among all folds, the best hyperparameters (i.e., the hyperparameter that resulted in higher accuracy, sensitivity, and specificity) are mtry = 4 for the nonfiltered peaks and mtry = 5 for the Ig-filtered peaks. It is worth mentioning that K-nearest neighbors, SVM-P, NB, and support vector machine radial (SVM-R) also had good performances, whereas NNET scored poorly. The ROC and PR curves for Ig-filtered and nonfiltered peaks, together with the mean accuracy, mean sensitivity, and mean specificity of the predictions by the six models, are presented in Fig 3C and D. The filtering by information gain (Ig) demonstrated a slightly worse performance when compared with the modeling without Ig filtering.

### Biomarker identification

Next, we focused on the identification of specific biomarkers based on the MALDI-TOF profile obtained. Within the MS features with the highest discriminatory value, statistical rank, and relative "m/z" peak intensity, a cluster of signals in the "m/z" 5,700–5,900 and 11,300–11,700 regions was chosen as a specific signature able to distinguish high- from low-risk COVID-19 patients (Fig 4A–D). Interestingly, the peaks at "m/z" 5,696, 5,724, 5,739, 5,765, 5,818, and 5,843 correspond to the doubly charged ions of "m/z" 11,393, 11,443, 11,476, 11,530, 11,633, and 11,683, respectively. This indicates that one or more proteoforms are contributing to discriminating high- and low-risk patients. To identify these proteins, the C18-fractionated plasma proteins/peptides were separated using 1D SDS–PAGE (Fig 5A and B). The "m/z" 10,000–15,000 region of the gel was excised, in-gel digested, and analyzed using nanoflow LC-MS/MS followed by data analysis. Quantitative proteomic analysis allowed the identification of 179 proteins with at least one unique peptide (Tables S6 and S7). Serum albumin, serotransferrin, complement C3a, *α*-2 macroglobulin, and haptoglobin were among the proteins with the

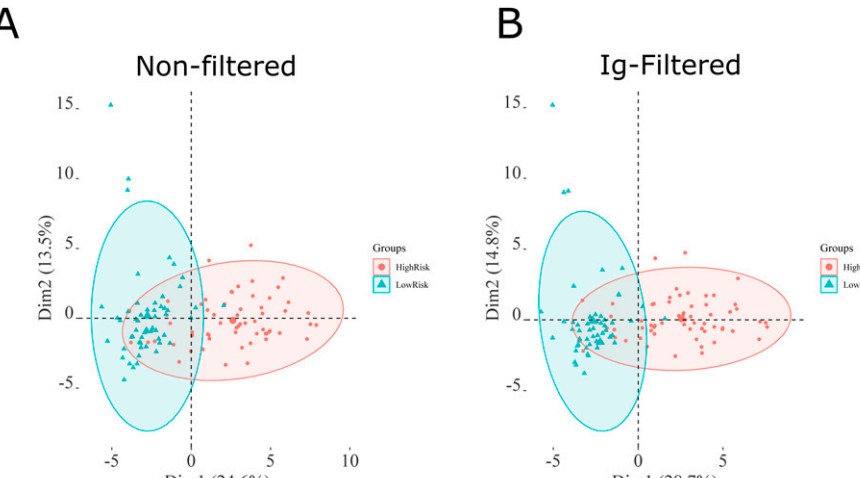

**Figure 2. PCA analysis of the preprocessed MALDI-TOF MS spectra obtained from 117 plasma samples.** **(A)** PCA of all significant peaks. **(B)** the PCA of peaks selected with the Ig (Information gain) method. The list of peaks is available in the Table S2.

highest PSMs. These proteins have molecular weights (MWs) higher than 15 kD, which is the MW cutoff of the gel band analyzed. A total of 52 proteins were identified with an MW between 10 and 15 kD. Within them, four proteins, platelet factor 4, immunoglobulin lambda variable 4-69 (IGLV4-69), serum amyloid A-1 (SAA1), and serum amyloid A-2 (SAA2), were up-regulated in the high- compared with low-risk group. The MALDI-TOF MS peaks at "m/z" 11,443, 11,530, and 11,683 correspond to truncated fragments originating from serum amyloid A1 (Table 3). The MALDI-TOF peaks at "m/z" 11,633, 11,476, and 11,393 correspond to truncated fragments originating from serum amyloid A2 (Table 3). The peptide [20]SFFSFLGEAFDGAR[33] peptide, shared between SAA1 and SAA2, was identified in the LC-MS/MS analysis and constitutes the initial sequence of one truncated form. The R19 was cleaved by trypsin during processing and constitutes the first amino acid of the other truncated form. The [21]FFSFLGEAFDGAR[33] semi-tryptic peptide, shared between SAA1 and SAA2, was also identified and constitutes the initial sequence of the other truncated form (Table S7). Moreover, the semi-tryptic peptides [23]SFLGEAFDGAR[33] and [24]FLGEAFDGAR[33] were identified, suggesting the presence of another two less-abundant proteo-forms, which were not detected in the MALDI-TOF MS spectra. These sequences were consistent with the MWs of the discriminatory "m/z" values identified up-regulated in the plasma samples collected from COVID-19 patients with high risk.

## Discussion

We describe the application of MALDI-TOF MS to identify a protein signature specific to COVID-19 patients with high and low risk, based on clinical symptoms, using 1 μl of C18-fractionated plasma. This study was based on the supposition that SARS-CoV-2 infection induces a systemic response that changes selectively the plasma protein expression, allowing a discrimination between patients at high risk (need of hospitalization) compared with low-risk ones (outpatient treatment). Using machine learning algorithms, the performance achieved a mean accuracy of 92%, a mean sensitivity of 93%, and a mean specificity of 92% was achieved separating the

two groups without feature selection. The sample preparation, data acquisition, and analysis parameters were optimized and validated to understand their influence of these factors in creating systemic biases. We confirmed that these factors were not influencing the accuracy of our approach based on the CVs detected. CVs reported in this study confirm similar reports from other research groups (17, 23, 24). In this study, we were interested in determining a specific protein or a panel of proteins that could be used for COVID-19 prognosis. Applying a combination of gel electrophoresis and nLC-MS/MS, we identified SAA1 and SAA2 proteoforms as regulated discriminatory proteins. These two proteins are involved in the acute phase response. Proteins involved in the acute phase response are increased early during viral and bacterial infections. Serum amyloid A-1 (SAA1) and A-2 (SAA2) are acute phase reactants synthesized by the liver and secreted into the bloodstream inflammatory and oncogenic processes (25). Extra-hepatic SAA protein synthesis has been reported in inflamed tissues (26, 27). SAA represents a family of high-density lipoproteins with 103–104 amino acids sharing high sequence homology between the different members. Four isoforms are expressed in humans SAA1, SAA2, SAA3, and SAA4 (25, 28). During infection, SAA protein production and secretion in the circulation can increase more than 1,000-fold suggesting an early response to infection. However, sustained expression of SAA proteins is associated with chronic pathological conditions.

SAA1 was already reported to be differentially expressed in patients with severe (29, 30). The SAA1 and SAA2 proteins were also identified up-regulated in severe COVID-19 patients in a clinical cohort from China (31). The authors used large-scale LC-MS/MS analysis of serum samples to identified differentially regulated proteins and metabolites as potential prognostic markers (31). The identification of SAA1 and SAA2 as potential markers confirms our study.

A specific correlation between SAA proteins and CRP has been found in several infectious diseases with the concentration of SAA increasing up to 2,000 mg/l (32). However, SAA proteins were found to be more sensitive than CRP in detecting variation in the inflammatory status of infected patients (33).

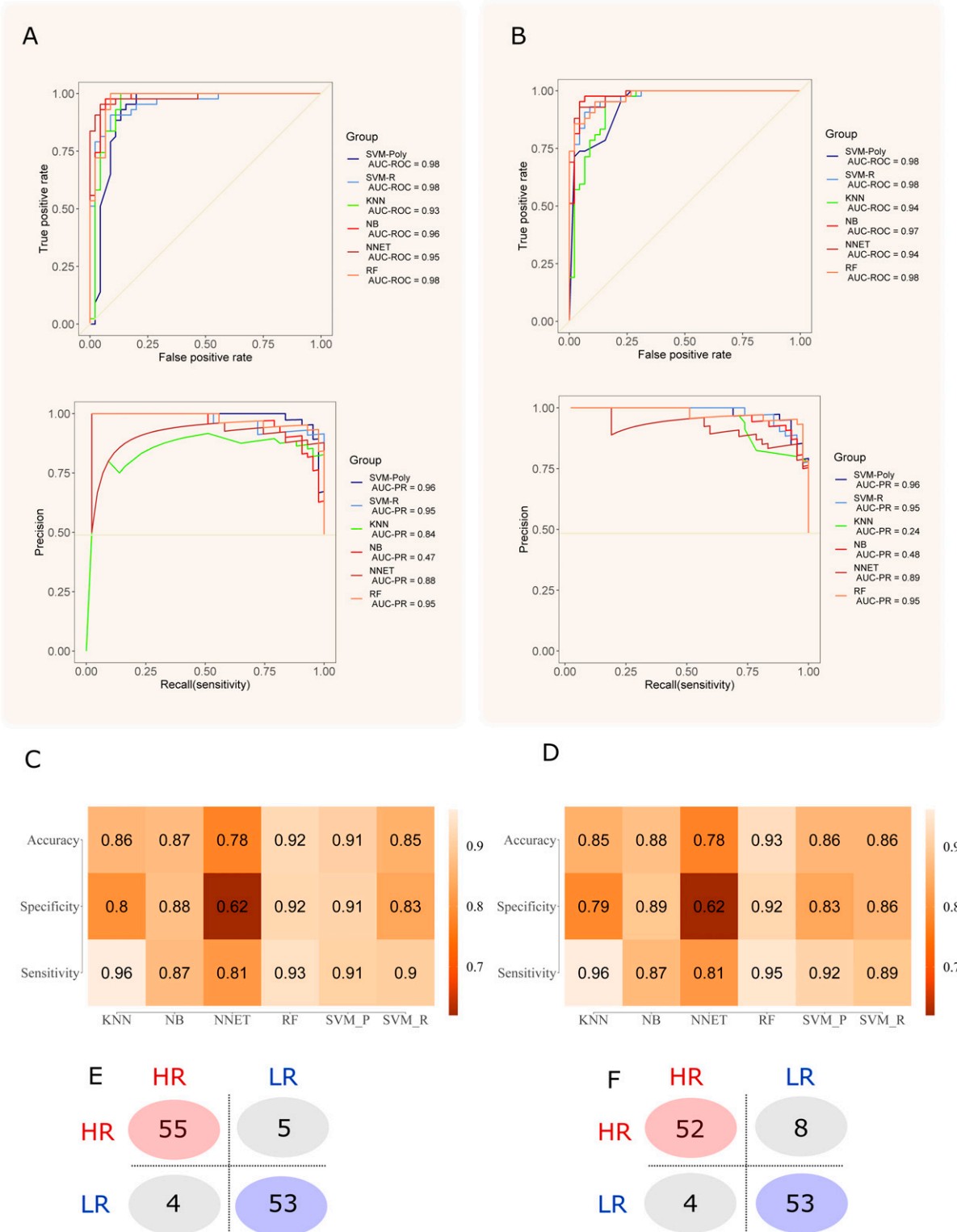

**Figure 3. Fourfold nested cross-validation of the dataset for model performance evaluation.**
**(A)** best AUC values for ROC and PR curves of each model with the non-filtered dataset. **(B)** best AUC values for ROC and PR curves of each model with the Ig-filtered dataset. **(C)** the average accuracy, specificity and sensibility of all folds for non-filtered peaks. **(D)** the average accuracy, specificity and sensibility of all folds for Ig-filtered peaks. **(E)** the total confusion matrix of the best model (Random Forest) for the non-filtered dataset. **(F)** the total confusion matrix of the best model (Random Forest) for the Ig-filtered dataset.

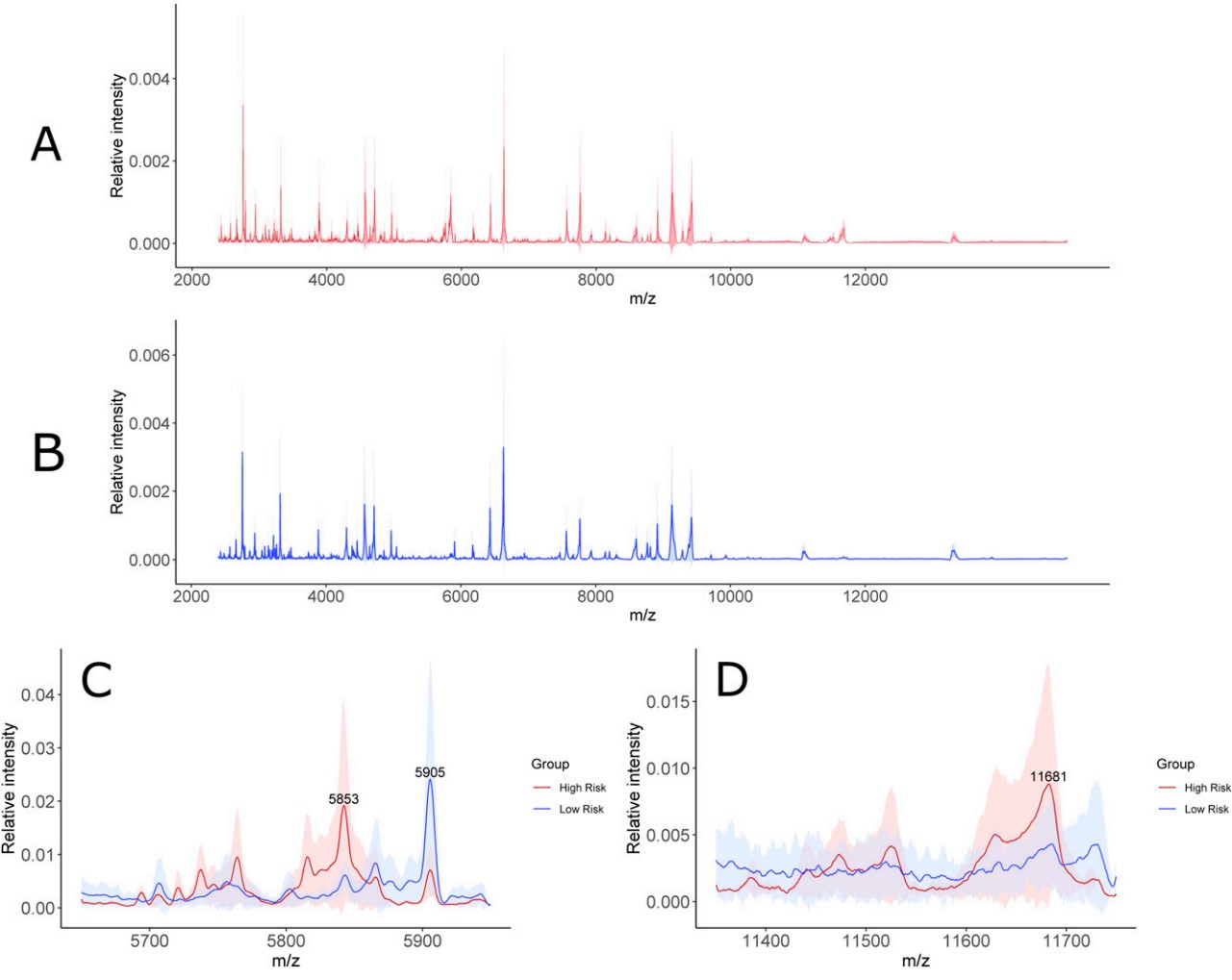

**Figure 4. Assessment of sample variability within the low and high-risk groups.**
**(A)** Average mass spectra with the interquartile range (IQR) shaded in red for the high-risk group (A). **(B)** Average mass spectra with the IQR shaded in blue for the low-risk group (B). **(C)** Comparison of the average mass spectra with the IQR shaded between the two groups for the most relevant discriminative peaks of "m/z 5,853–5,905 (C). **(D)** Comparison of the average mass spectra with the IQR shaded between the two groups for the most relevant discriminative peak at "m/z" 11,681 (D).

Because of that, increased levels of SAA1 and SAA2 proteoforms can be seen as a measure of the increased severity of the disease and so on prognostic factors. Because of its ubiquitous expression in several infectious diseases, SAA proteins cannot be associated directly with the SARS-CoV-2 and should be complemented with other viral specific molecular tests.

A possible mechanism of increase in SAA proteins in severe COVID-19 patients could be due to the cytokine storm that is elicited during the infection. Indeed, increased levels of cytokines such as interleukin IL-2, IL-7, granulocyte (G)CSF, interferon-γ inducible protein 10, MCP 1, MIP 1-α, and TNF-α and IL-6 is associated with COVID-19 disease severity, suggesting that the mortality observed could be due to virally/induced hyperinflammation (34). The elevation of IL-1 and IL-6 increase synergistically the levels of SAA proteins synergistically. At the same time, SAA proteins increase the expression of IL-1β mediated by NLRP3 in human and mouse immune cells (35, 36). SARS-CoV

ORF8b activates the NLRP3 inflammasome inducing the secretion of active IL-1β and IL-18 (37). Moreover, SARS-CoV ORF3a activates the NLRP3 inflammasome by promoting TNF receptor–associated factor 3 (TRAF3) ubiquitination of p105 and activation of NF-kB and subsequent transcription and secretion of IL-1β (38). Overactivation of NLRP3 in SARS-CoV-2 infection has been postulated delineating specific pathways for its activation (38, 39, 40). Blockade of NFκB, a central player in the SAA-mediated activation of proinflammatory cytokines could represent a novel therapeutic target for severe cases of COVID-19. Because of that, SAA proteins might play a critical role in SARS-CoV-2 infection as an early response to inflammation but also can be seen as proinflammatory proteins to amplify the cytokine storm. Although comprehensive LC-MS/MS analysis has been performed using sera from COVID-19 patients, a proteomic fingerprint using MALDI-TOF MS on plasma samples has not been reported. Recently, MALDI-TOF MS combined with a machine learning

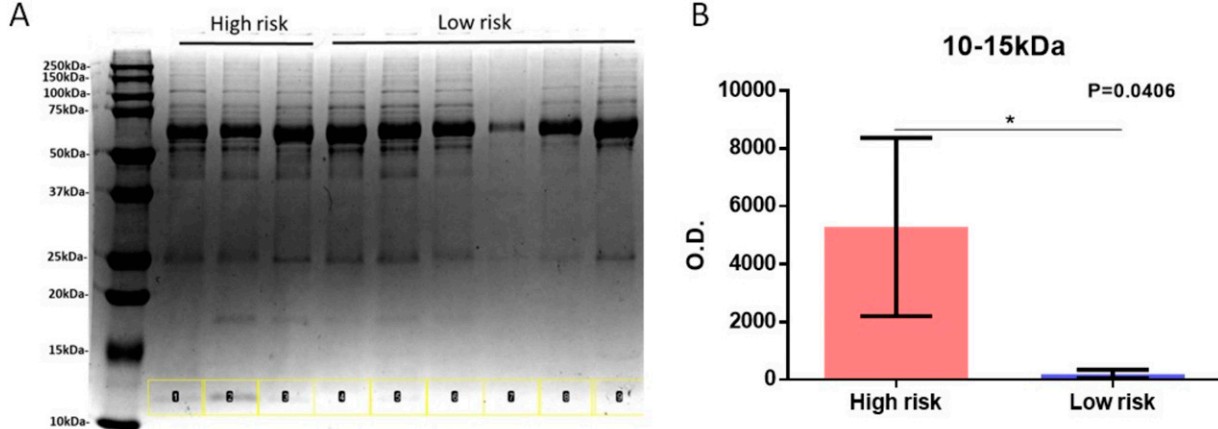

**Figure 5. 1D-SDS–PAGE of C18-fractionated plasma from high- (3) and low- (6) risk patients.**
**(A)** The region between 10 and 15 kD is highlighted in yellow. **(B)** Quantification of the 10–15 kD region in the high and low-risk groups.

approach was used to detect SARS-CoV-2 in nasal swabs from infected patients (41). The application of RT-PCR, immunochromatography, and recently MALDI-TOF MS has been used and proven reliable for the diagnosis of SARS-CoV-2 infection. However, no method exists so far to discriminate between high- and low-risk patients. This study shows that MALDI-TOF MS combined with machine learning algorithms offers a reproducible, easy-to-use, fast, low-cost technique that can be implemented by several researchers worldwide to test the reliability of this marker. Moreover, the widespread use of MALDI-TOF in clinical laboratories will allow an easy transition into the hospitals.

### Limitations of the study

This study has focused on the fractionated plasma focusing on a limited mass range "m/z" 2,000–20,000. Moreover, the concomitant ionization of proteins/peptides in this region limits the detection of low abundant ones. Improved large-scale shotgun approaches combined with extensive fractionation have been applied to identify potential COVID-19 biomarkers and could be used in association with SAA1 and SAA2 provided in this study to create a panel of more reliable biomarkers. Association of the current biomarkers with other biomarkers will offer the possibility to improve the prognostic accuracy. Further validation in prospectively

collected samples and proof of benefit to the existing noninvasive diagnostic strategies are required.

A larger independent cohort of patients should be analyzed to corroborate these findings. Inter-laboratory studies across countries should be performed to validate these data. Moreover, a time-course study during the development of the infection would give more information on the validity of these markers as early prognostic markers.

## Materials and Methods

### Study subjects and design

Plasma from a total of 117 patients with COVID-19 divided into high risk (n = 57) and low risk (n = 60) was collected prospectively from a Brazilian cohort (Tables 1 and 2) at the Heart Institute (InCor) and Central Institute, University of São Paulo Medical School, Brazil, between March, 2020 and July, 2020 in consecutive sampling. The study was approved according to the principles expressed in Declaration of Helsinki by Comissão nacional de ética em pesquisa and local Ethics Committees (CAAE 30299620.7.0000.0068). All patients signed an informed consent form. Patients with high risk were defined based on clinical parameters evaluated at the time of admission that required hospitalization compared with low-risk

**Table 3. Sequences of truncated serum amyloid protein A-1 and A-2 identified as discriminant peaks in the MALDI-TOF MS analysis and sequenced using nLC-MS/MS.**

| Protein name | Sequence | MW, experimental (D) | MW, theoretical (D) |
|---|---|---|---|
| Serum amyloid A-1 (SAA1) | [19]RSFFSFLGEAFDGARDMWRAYSD—AGLPEKY[122] | 11,683 | 11,675.49 |
| Serum amyloid A-1 (SAA1) | [20]SFFSFLGEAFDGARDMWRAYSD—AGLPEKY[122] | 11,530 | 11,519.39 |
| Serum amyloid A-1 (SAA1) | [21]FFSFLGEAFDGARDMWRAYSD—AGLPEKY[122] | 11,443 | 11,432.00 |
| Serum amyloid A-2 (SAA2) | [19]RSFFSFLGEAFDGARDMWRAYSD—AGLPEKY[122] | 11,633 | 11,640.60 |
| Serum amyloid A-2 (SAA2) | [20]SFFSFLGEAFDGARDMWRAYSD—AGLPEKY[122] | 11,476 | 11,484.50 |
| Serum amyloid A-2 (SAA2) | [21]FFSFLGEAFDGARDMWRAYSD—AGLPEKY[122] | 11,393 | 11,397.47 |

patients. Cases were included with a clinical picture suggestive of COVID-19 defined as two or more of the following: cough, fever, shortness of breath, diarrhea, myalgia, headache, sore throat, running nose, sudden gustatory or olfactory loss, and detection of viral RNA in nasopharyngeal SARS-CoV-2 PCR positive. Patients with high and low risk of hospitalization were matched for confounding variables such as age, sex, and comorbidities to explain the difference between groups (Table S1). Plasma samples were collected, aliquoted and stored at –80°C for further analyses.

### Sample preparation for MALDI-TOF MS analysis and data processing

Venous punctures from the patients were performed. After the samples were collected into tubes containing EDTA anticoagulant, these were centrifuged in a refrigerated unit at 5,000*g* for 15 min at 4°C. Then, samples were carefully removed from the centrifuge not to resuspend cells, and the plasma fraction was collected and aliquoted at –80°C until further analyses.

Different sample preparation strategies were evaluated for profiling the plasma proteome of COVID-19 patients. (1) Thawed plasma samples were diluted 1:100 in water. Matrix solution (sinapinic acid [SA], dihydroxybenzoic acid [DHB], and *α*-cyano-hydroxycinnamic acid [HCCA]) were prepared by dissolving in acetonitrile/water 50:50 vol/vol containing 0.1% or 2.5% TFA at 10 mg/ml and was mixed with 1 *μ*L of diluted serum and directly spotted in duplicate onto a stainless steel MALDI target plate (Bruker Daltonics). (2) C18-based plasma protein extraction. C18 polymeric disks were inserted into p200 pipette tips to produce a micro-column. The disks were activated with 100 *μ*l 100% methanol and conditioned with 0.1% TFA. 1 *μ*l of plasma samples was diluted 1:10 in 0.1% TFA and further acidified to achieve 1% TFA. After acidification samples were spun down at 10,000*g* for 10 min and the supernatant loaded into the micro-column. The column was further washed with 100 *μ*l of 0.1% TFA and proteins eluted with a matrix directly onto the MALDI plate. All steps except the elution were performed in a bench centrifuge at 1,000*g* for 2 min to improve sample processing and reproducibility of the entire strategy.

Samples were analyzed in a MALDI-TOF Autoflex speed smart-beam mass spectrometer (Bruker Daltonics) using FlexControl software (version 3.3; Bruker Daltonics). Spectra were recorded in the positive linear mode (laser frequency, 500 Hz; extraction delay time, 390 ns; ion source 1 voltage, 19.5 kV; ion source 2 voltage, 18.4 kV; lens voltage, 8.5 kV; mass range, 2,400–20,000 D). Spectra were acquired using the automatic run mode to avoid subjective interference with the data acquisition. For each sample, 2,500 shots, in 500-shot steps, were summed. All spectra were calibrated by using Protein Calibration Standard I (Insulin $[M+H]^+$ = 5,734.52, Cytochrome C $[M+2H]^{2+}$ = 6,181.05, Myoglobin $[M+2H]^{2+}$ = 8,476.66, Ubiquitin I $[M+H]^+$ = 8,565.76, Cytochrome C $[M+H]^+$ = 12,360.97, Myoglobin $[M+H]^+$ = 16,952.31) (Bruker Daltonics).

The data preprocessing was performed using the ClinProTools, FlexAnalysis 4.0 (Bruker Daltonics), and R-packages. The ClinProTools software was used for MS spectra visualization and R-packages for data processing. The pipeline for processing the raw files and applying the models was adapted from reference 16. Fid files were converted to mzML using the MSconvert function from the

ProteoWizard suit (version: 3.0.20220) (17). Then, the files were preprocessed using MALDIquant and MALDIquantForeign packages (18). The spectra range was trimmed (2.5–15 kD). The resulting files were transformed (square root method) and smoothed (Savitzky–Golay method), and the baseline correction was done by the TopHat algorithm (19, 20). Intensities of all files were normalized (total ion current calibration method), and the peaks were detected with a signal-to-ratio noise of two and a half-WindowSize of 10 (16). For each group, peaks were binned with a tolerance of 0.003, keeping the ones present in 80% of the samples; next, the peaks of both groups were binned together. Sample normality was accessed by a Shapiro–Wilk test and a two-tailed Wilcoxon rank sum test corrected for multiple hypotheses testing using the Benjamini–Hochberg was performed. A significant difference was considered for *P*-values < 0.05. To evaluate If the observed differences were simply by chance, we permuted the dataset 100 times and calculated the global false discovery rate. The resultant dataset was used for the PCA analysis and the machine learning analysis. In addition, peaks were filtered using the information gain (Ig) function of the FSelector package to search for the most relevant features, this method was used because it is classifier independent and is faster than wrapper methods, which is desirable when comparing multiple machine learning algorithms (21). Features with a weight higher than 0 were used for PCA and machine learning analysis.

### Machine learning

For peaks with and without Ig filtering, six different algorithms (SVM-P, SVM-R, KNN, neural net [NNET], NB, and RF) were accessed to classify high- and low-risk samples. To choose between the models, the training and testing were performed through fourfold nested repeated five times 10-fold cross-validation using the Caret package in R, the data were split randomly into the folds (16, 22). For hyperparameter optimization, a random search among 10 parameters was performed in the inner loop. ROC and PR curves were created using the MLeval package. The AUC and PR curves from the best results were reported. Also, the mean accuracy, sensitivity, and specificity metrics for the cross-validation predictions were calculated.

### 1D SDS–PAGE and nanoflow liquid chromatography coupled to tandem MS analysis

Proteins from case and control samples were separated by one-dimensional gel electrophoresis using a 12% gel. Gels were stained using Coomassie brilliant blue and the gel was scanned to identify differentially expressed bands. Bands were excised in the MW range of 10,000–15,000 D corresponding to the "m/z" of discriminant peaks. Bands were in-gel tryptic digested according to Shevchenko and subjected to nanoflow LC-MS/MS analysis. The nLC-MS/MS analysis was performed using an Easy nano LC1000 (Thermo Fisher Scientific) HPLC coupled with an LTQ Orbitrap Velos (Thermo Fisher Scientific). Peptides were loaded on a C18 EASY-column (2 cm × 5 × 100 *μ*m; 120 Å pore; Thermo Fisher Scientific) using a 300 nl/min flow rate of mobile phase A (0.1% formic acid) and separated in a C18 PicoFrit PepMap (10 cm × 10 × 75 *μ*m; 135 Å pore; New Objective), over 105 min using a linear gradient 2–30% of mobile phase B (100% ACN; 0.1% formic acid).

The eluted peptides were ionized using electrospray. The top 20 most intense precursor ions with charge-state ≥ 2 were fragmented using collision-induced dissociation at 35 normalized collision energy and 10 ms activation time. The MS scan range was set between "m/z" 350–1,500, the MS scan resolution was 60,000, the MS1 ion count was $1 \times 10^6$ and the MS2 ion count was $3 \times 10^4$.

### Statistical analysis, database search, and quantitative analysis

nLC-MS/MS raw data were searched using Proteome Discoverer (v2.3.0.498; Thermo Fisher Scientific) for protein identification and label-free quantification quantification. The raw files were searched against the *Homo sapiens* protein database containing 20,359 reviewed protein sequences (UniProt, downloaded in June, 2020). The database search was performed using the Sequest HT processing node with trypsin semi-specific as the proteolytic enzyme, two missed cleavages, 10 ppm precursor ion tolerance, and 0.6 D fragment ions mass tolerance. Carbamidomethylation of cysteine was set as fixed modification and methionine oxidation as dynamic modification. Label-free quantification was performed using the Minora algorithm in the processing workflow embedded in Proteome Discoverer 2.3. Precursor Ions Quantifier node and the Feature Mapper were added to the consensus workflow for retention time alignment.

### Patient and public involvement

This study analyzed a retrospective case-series cohort. No patients were involved in the study design, setting the research questions, or the outcome measures directly. No patients were asked to give advice on interpretation or writing up of results.

## Data Availability

MALDI-TOF MS spectra and the patients' categories were uploaded in the PRIDE public repository (https://www.ebi.ac.uk/pride/), dataset identifier PXD025138, Username: reviewer_pxd025138@ebi.ac.uk, Password: WEvrGIzv. LC-MS/MS data were submitted to PRIDE (https://www.ebi.ac.uk/pride/), project number PXD021581, Username: reviewer_pxd021581@ebi.ac.uk, Password: 79CZBtm6.

## Supplementary Information

## Acknowledgements

This work was supported by Fundação de Amparo à Pesquisa do Estado de São Paulo (FAPESP), G Palmisano (2018/18257-1, 2018/15549-1, 2020/04923-0), C Wrenger (2015/26722-8, 2017/03966-4), CRF Marinho (2018/20468-0), and JC Nicolau (2020/04705-2). G Palmisano, C Wrenger, and CRF Marinho were supported by Conselho Nacional de Desenvolvimento Científico e Tecnológico (CNPq). The funders had no role in study design. L Rosa-Fernandes and VF Santiago are supported by: Coordenação de Aperfeiçoamento de Pessoal de Nível Superior.

## Author Contributions

LC Lazari: data curation, software, formal analysis, validation, investigation, visualization, methodology, and writing—original draft, review, and editing.
FDR Ghilardi: resources, data curation, investigation, visualization, methodology, and writing—original draft, review, and editing.
L Rosa-Fernandes: data curation, formal analysis, investigation, visualization, methodology, and writing—review and editing.
DM Assis: data curation, formal analysis, investigation, visualization, methodology, and writing—original draft, review, and editing.
JC Nicolau: resources, data curation, supervision, investigation, methodology, project administration, and writing—review and editing.
VF Santiago: data curation, investigation, visualization, methodology, and writing—review and editing.
TF Dalçóquio: resources, data curation, investigation, methodology, and writing—review and editing.
CB Angeli: data curation, investigation, visualization, methodology, and writing—review and editing.
AJ Bertolin: resources, data curation, investigation, methodology, and writing—review and editing.
CRF Marinho: resources, data curation, investigation, methodology, and writing—review and editing.
C Wrenger: resources, data curation, investigation, methodology, and writing—review and editing.
EL Durigon: resources, data curation, investigation, methodology, and writing—review and editing.
RF Siciliano: resources, data curation, formal analysis, supervision, funding acquisition, validation, investigation, visualization, methodology, project administration, and writing—original draft, review, and editing.
G Palmisano: conceptualization, resources, data curation, software, formal analysis, supervision, funding acquisition, validation, investigation, visualization, methodology, project administration, and writing—original draft, review, and editing.

### Conflict of Interest Statement

The authors declare that they have no conflict of interest.

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
