## [Reviewer comments · Life Science Alliance]

Life Science Alliance

Prognostic accuracy of MALDI-TOF mass spectrometric analysis of plasma in COVID-19

Lucas Lazari, Fabio Ghilardi, Livia Rosa-Fernandes, Diego Assis, Jose Nicolau, Veronica Santiago, Talia Dalçóquio, Claudia Angeli, Adriadne Bertolin, Cláudio Marinho, Carsten Wrenger, Edison Luiz Durigon, Rinaldo Siciliano, and Giuseppe Palmisano

DOI: <https://doi.org/10.26508/lsa.202000946>

Corresponding author(s): Giuseppe Palmisano, University of Sao Paulo

Review Timeline:	Submission Date:	2020-10-23
	Editorial Decision:	2021-01-05
	Revision Received:	2021-04-01
	Editorial Decision:	2021-05-18
	Revision Received:	2021-05-31
	Accepted:	2021-05-31

Scientific Editor: Shachi Bhatt

Transaction Report:

January 5, 2021

Re: Life Science Alliance manuscript #LSA-2020-00946-T

Prof Giuseppe Palmisano
University of Sao Paulo
Department of Parasitology
Avenida Lineu Prestes 1374
Sao Paulo, Sao Paulo 05508-000
Brazil

Dear Dr. Palmisano,

Thank you for submitting your manuscript entitled "Prognostic accuracy of MALDI-TOF mass spectrometric analysis of plasma in COVID-19" to Life Science Alliance. The manuscript was assessed by expert reviewers, whose comments are appended to this letter.

As you can see from the reviewers' comments, while the reviewers are enthusiastic about the findings, they have pointed out a number of errors that should be addressed prior to further consideration of the manuscript. We this invite to submit a revised manuscript that addresses all the concerns of the reviewers.

Thank you for this interesting contribution to Life Science Alliance. We are looking forward to receiving your revised manuscript.

Sincerely,

Shachi Bhatt, Ph.D.
Executive Editor
Life Science Alliance
<https://www.lsjournal.org/>
Tweet @SciBhatt @LSAJournal

- A letter addressing the reviewers' comments point by point.
- An editable version of the final text (.DOC or .DOCX) is needed for copyediting (no PDFs).
- High-resolution figure, supplementary figure and video files uploaded as individual files: See our detailed guidelines for preparing your production-ready images, <https://www.life-science-alliance.org/authors>
- Summary blurb (enter in submission system): A short text summarizing in a single sentence the study (max. 200 characters including spaces). This text is used in conjunction with the titles of papers, hence should be informative and complementary to the title and running title. It should describe the context and significance of the findings for a general readership; it should be written in the present tense and refer to the work in the third person. Author names should not be mentioned.

B. MANUSCRIPT ORGANIZATION AND FORMATTING:

Reviewer #1 (Comments to the Authors (Required)):

This presented manuscript is in context of today's SARS-CoV-2 corona pandemic of highest scientific and particular of pharmaceutical and medical interest. The manuscript is most well prepared and written, comprehensible, technically sound and certainly of high significance for the international scientific community, as well as of high interest for the general readership of the journal.

All experiments described were well performed and are conclusive, applying complementary advanced MS methods, extended to machine learning. The overall interpretation and conclusion are

also well presented, considering also limitations of the present experiments, in context of the cohort of patient selected and available sample collection.

The paper certainly deserves rapid publication.

Minor comments:

The authors may consider to bring in the abstract a brief introduction for serum amyloid A1 and A2, ... known already to be biomarkers for infections...

Section Results

Extend the sentence, 2nd line... method to investigate plasma samples
2) allowing a clinical application for plasma isolated....

Discussion

The authors need to cross-check reference 31, and its position

Limitations of the study

1. line mass range 2000-20000 , unit missing

Table 1

Time onset symptoms, units are days ?

Table 2 and 3

Last column, p needs to be explained in the legend

Reviewer #2 (Comments to the Authors (Required)):

There is a critical need for new methods in the prognosis of COVID-19. Palmisano describes plasma proteome fingerprint to predict high-risk and low-risk cases of COVID-19 identified by a previously described platform that combines machine learning with matrix-assisted laser desorption ionization mass spectrometry (MALDI-TOF MS) analysis. A good accuracy (93%), sensitivity (87.5%) and specificity (100%) was proposed in the blinded test.

1. Abstract. The statement "cases of COVID-19 identified by a novel platform that combines machine learning with matrix-assisted laser desorption ionization mass spectrometry (MALDI-TOF MS) analysis" is not correct at all. The first work combining ML and MALDI-MS for COVID was the cited reference 16. Thus, abstract must be changed accordingly.
2. The specificity is not 100% in Figure 3C/D. Correct it.
3. Page 3, correct the phrase: "peaks at 16616.3, 13315.1, 11095.6, 9496.2 and 8316.8 m/z corresponded" to "peaks at m/z 16616.3, 13315.1, 11095.6, 9496.2 and 8316.8 corresponded"
4. Page 3, correct: "in the 6000 m/z" to "in the m/z 6000"
5. Page 3, correct: "from 2000-10000 m/z" to "from m/z 2000-10000"
6. Please correct throughout the text the way to describe ions as "m/z XXXX" and not "XXXX m/z."
7. Page 3, correct: 1 μ L
8. Page 4, correct: "chi-square test $p < 0,005$ " to "chi-square test $p < 0.005$ "

9. Page 5: the description of the ions changed from m/z to Da. Correct it.
10. Page 5: 15 kDa; 10-15 kDa
11. Throughout the text numbers and units are described wrong. The correct manner to describe numbers and units must have a space between them: 10 mg/L; 1 μ L; etc
12. Figure 5 should be converted to Centroid spectra.
13. References must be updated - many of them have page numbers and the authors describe DOI numbers
14. The Wilcoxon test should include multiple hypothesis correction (FDR as example).
15. Figure 5: The figure should ideally include information about the variability of the intensities across the samples as well, for example, by shading the IQRs. It is necessary to be added the variance to the mean spectrum of all High versus Low-risk controls in figure 5. This is fundamental to understanding the variance observed and if the differences in the peaks are reliable. This is analogous to providing the box-and-whisker plot for all the values in the spectrum. Comparison of the intensities the most relevant peaks to differentiate the High-risk group from the Low-risk group - data should be presented as mean values {plus minus} IQR.
16. Why was lg used as Feature Selection method? This description in the text is confusing and no reason is given, only that was adapted from reference 16 (Nat. Biotech 2020).
17. I have some questions about how the machine learning models were trained. Performing lg selection as part of the model training pipeline is an important improvement, but performance of the models might still be overestimated due to the way the cross validation is performed. To train the initial model, it is mentioned that 4-fold cross validation was performed 10 times for all model and lg combinations??? A nested cross validation was used for hyperparameter tuning. Please clarify why this is not the case and how this single list of selected features was obtained (when during model training, which samples).
18. It is suggested that the authors first employ Feature Selection (lg) and attempt to control for FDR in the error rates. Permutation of the samples labels (High and Low risk patients) and recalculation of the statistical differences would provide some indication if the observed differences are simply due to chance. See a method in (<https://www.ncbi.nlm.nih.gov/pmc/articles/PMC3501526/figure/pone-0049724-g003/>) and cited reference 16 (Nachtigall, Nat. Biotech. 2020)
19. Page 4 it is described that lg filtering demonstrated a better separation than the PCA. Was it used only 5 peaks in the lg analysis?
20. The evaluations of machine learning is another aspect that warrants more information, careful re-assessments, and/or improved clarity. It is hopeful that there is a real signal in the ML, but the machine learning requires more scrutiny. For example, the "validation" on page 4 isn't a true external validation, the model is just retrained on a different dataset. This suffers from the same issues as mentioned above and thus doesn't provide any additional insight in the validity and generalizability of the initial model. If the method was robust and the model was generalizable, no lg or retraining would need to be required to get good predictions. Additionally, for this evaluation there is even less information available: how many features were retained after lg, ...? Please describe better.
21. There is a concern about data leakage as based on the description provided the machine learning was not quite done correctly. For example, the lg is performed on all the data prior to splitting the data into train and test sets for training the machine learning models. This will lead to information the data into train and test sets for training the machine learning models. This will lead to information leakage to the test set, resulting in optimistically biased model performance. Suggested solution: lg needs to be integrated in the model evaluation and the whole pipeline needs to then be properly evaluated on a test set. It could also be that we misinterpreted the description of the methods, and in the absence of the code and the data input this could not be verified. If it was a misinterpretation then clarify and provide the data tables and code.

22. How does the 10-fold cross-validation samples ordered for evaluation? Were the 10-fold cross-validation with the COVID-19 samples ordered first and the healthy samples ordered second? Explain. In that case for the first few folds the test set will only contain COVID-19 samples and the final few folds will only contain healthy samples. Thus, the evaluation will be completely biased towards one of the two classes. As a suggestion for the authors: randomized stratified splitting should be used.

23. Please provide hyperparameters of the machine learning models.

24. Data and code availability: The SI is unacceptable and not compatible with a good magazine quality. There is no FID data from MALDI-MS spectra. At a minimum they should be the exl, csv or tab-delimited file so it can be efficiently reused. It should need a description of what the SI is in the first place. The data from this study has the potential to become very useful for people throughout the world and should be deposited publicly in a proteomic or metabolomics repository or generalist repositories and can get its own DOI in several such repositories for the data set. Ideally the data is in both the raw format and open format. If data is not allowed to be made available provide explicit reasons. All new code needs to be accessible, the data tables to make the figs and the scripts to create them. Also have the authors considered creating a systematic web to link data information about the data where this is stored in public repositories where people throughout the world can then compare their data to everyone else's data, including for this study. I envision there will be a lot of interest in this from the wider diagnostic community as the MALDI systems are so widely available. One way may be through the PRIDE repository as they have make such data available and accessible.

I want to iterate that the potential of the tools can be transformative and that I am enthusiastic about the approach but that it will need a lot of points to address and there are hints that are very encouraging but the work needs more refinement. And if after the refinement the results still hold then this work should be published ASAP. I will be happy to rereview in an expedited manner the corrected version.

Reply to reviewers' comments

Reviewer #1 (Comments to the Authors (Required)):

This presented manuscript is in context of today's SARS-CoV-2 corona pandemic of highest scientific and particular of pharmaceutical and medical interest. The manuscript is most well prepared and written, comprehensible, technically sound and certainly of high significance for the international scientific community, as well as of high interest for the general readership of the journal.

All experiments described were well performed and are conclusive, applying complementary advanced MS methods, extended to machine learning. The overall interpretation and conclusion are also well presented, considering also limitations of the present experiments, in context of the cohort of patient selected and available sample collection.

The paper certainly deserves rapid publication.

We would like to thank Reviewer #1 for the positive and important comments that have helped us to improve the manuscript. We have replied to all reviewers' comments.

Moreover, we have applied the method developed in this manuscript to another blind dataset showing high accuracy, sensitivity and specificity confirming the validity of the method. We have deposited the LC-MS/MS and MALDI-MS data to public repositories and made available the code. Since Life Science Alliance journal allows a one cycle revision, we believe that the all the improvements implemented in the revised version of the manuscript makes it suitable for publication in Life Science Alliance journal.

Minor comments:

The authors may consider to bring in the abstract a brief introduction for serum amyloid A1 and A2, ... known already to be biomarkers for infections...

Added.

Section Results

Extend the sentence, 2nd line... method to investigate plasma samples
2) allowing a clinical application for plasma isolated....

Added.

Discussion

The authors need to cross-check reference 31, and its position
Changed the reference.

Limitations of the study

1. line mass range 2000-20000, unit missing

Added.

Table 1

Time onset symptoms, units are days ?

We added this information in the table.

Table 2 and 3

Last column, p needs to explained in the legend

The explanation for the pvalues has been added.

Reviewer #2 (Comments to the Authors (Required)):

There is a critical need for new methods in the prognosis of COVID-19. Palmisano describes plasma proteome fingerprint to predict high-risk and low-risk cases of COVID-19 identified by a previously described platform that combines machine learning with matrix-assisted laser desorption ionization mass spectrometry (MALDI-TOF MS) analysis. A good accuracy (93%), sensitivity (87.5%) and specificity (100%) was proposed in the blinded test.

We thank reviewer #2 for the positive and important comments that have helped us to improve the manuscript. We have replied to all reviewers' comments.

Moreover, we have applied the method developed in this manuscript to another blind dataset showing high accuracy, sensitivity and specificity confirming the validity of the method. The results of this analysis are available for reviewer purpose. We have deposited the LC-MS/MS and MALDI-MS data to public repositories and made available the code. Since Life Science Alliance journal allows a one cycle revision, we believe that the all the improvements implemented in the revised version of the manuscript makes it suitable for publication in Life Science Alliance journal.

1. Abstract. The statement "cases of COVID-19 identified by a novel platform that combines machine learning with matrix-assisted laser desorption ionization mass spectrometry (MALDI-TOF MS) analysis" is not correct at all. The first work combining ML and MALDI-MS for COVID was the cited reference 16. Thus, abstract must be changed accordingly.

Removed the word "novel".

2. The specificity is not 100% in Figure 3C/D. Correct it.

We have changed the Figure 3C and 3D.

3. Page 3, correct the phrase: "peaks at 16616.3, 13315.1, 11095.6, 9496.2 and 8316.8 m/z corresponded" to "peaks at m/z 16616.3, 13315.1, 11095.6, 9496.2 and 8316.8 corresponded"

Corrected.

4. Page 3, correct: "in the 6000 m/z" to "in the m/z 6000"

Corrected.

5. Page 3, correct: "from 2000-10000 m/z" to "from m/z 2000-10000"

Corrected.

6. Please correct throughout the text the way to describe ions as "m/z XXXX" and not "XXXX m/z."

Corrected.

7. Page 3, correct: 1 μ L

Corrected.

8. Page 4, correct: "chi-square test $p < 0,005$ " to "chi-square test $p < 0.005$ "

Corrected, now in page 3.

9. Page 5: the description of the ions changed from m/z to Da. Correct it.

Corrected.

10. Page 5: 15 kDa; 10-15 kDa

Corrected.

11. Throughout the text numbers and units are described wrong. The correct manner to describe numbers and units must have a space between them: 10 mg/L; 1 μ L; etc

Corrected.

12. Figure 5 should be converted to Centroid spectra.

We did not convert the spectra to centroid mode to better show the peak intensity variability for the high and low risk group. Moreover, Figure 5 (now Figure 4) was modified according to the reviewer's suggestion reported below including the shading representing the mean \pm IQR. A zoom in the peak area of some of the most representative discriminative peaks is shown.

13. References must be updated - many of them have page numbers and the authors describe DOI numbers

The references were updated.

14. The Wilcoxon test should include multiple hypothesis correction (FDR as example).

FDR was calculated after the Wilcoxon test using the Benjamini-Hochberg method for calculating the adjusted p values. This information was added in the text.

15. Figure 5: The figure should ideally include information about the ^[SEP]variability of the intensities across the samples as well, for example, ^[SEP]by shading the IQRs. ^[SEP]It is necessary to

be added the variance to the mean spectrum of all High versus Low-risk controls in figure 5. This is fundamental to understanding the variance observed and if the differences in the peaks are reliable. This is analogous to providing the box-and-whisker plot for all the values in the spectrum. Comparison of the intensities the most relevant peaks to differentiate the High-risk group from the Low-risk group - data should be presented as mean values {plus minus} IQR.

Figure 5 (now Figure 4) was modified according to the reviewer's suggestion reported below including the shading representing the mean±IQR. A zoom in the peak area of some of the most representative discriminative peaks is shown.

16. Why was Ig used as Feature Selection method? This description in the text is confusing and no reason is given, only that was adapted from reference 16 (Nat. Biotech 2020).

Ig was used because we wanted to test if the feature selection could improve our results, that is the main reason why we opted for this strategy. The selection for the Ig method was due to the fact that it is faster than the wrapper methods and is classifier independent, which is desirable when comparing multiple ML algorithms, as stated by ref 16.

This explanation was added in the methods section.

17. I have some questions about how the machine learning models were trained. Performing Ig selection as part of the model training pipeline is an important improvement, but performance of the models might still be overestimated due to the way the cross validation is performed. To train the initial model, it is mentioned that 4-fold cross validation was performed 10 times for all model and Ig combinations???. A nested cross validation was used for hyperparameter tuning. Please clarify why this is not the case and how this single list of selected features was obtained (when during model training, which samples).

We performed the 4-fold nested repeated Cross Validation (CV) in two datasets separately. First, we employed only the significant peaks in the six different models, since our random search for hyperparameters has a tuneLength = 10, the repeated 10-fold CV was performed 10 times to optimize parameters. Later, the same was done for the data set with peaks filtered by information gain and the results were compared.

The list of selected features was obtained within each fold. That is, we separated the data in 4-folds, then in the outer loop one fold was designated to be the test fold and the remaining were designated as the trainings fold. We performed the Ig selection in the training folds only, to avoid leaking information between the train and test. The model was then trained with the filtered dataset and then the untouched test set was applied to the model.

18. It is suggested that the authors first employ Feature Selection (Ig) and attempt to control for FDR in the error rates. Permutation of the samples labels (High and Low risk patients) and recalculation of the statistical differences would provide some indication if the observed differences are simply due to chance. See a method in (<https://www.ncbi.nlm.nih.gov/pmc/articles/PMC3501526/figure/pone-0049724-g003/>) and cited reference 16 (Nachtigall, Nat. Biotech. 2020)

We employed a permutation to calculate the global FDR. We tried to use the tool provided in the reference suggested, however it wasn't working. Thus, we permuted the samples 100

times and calculated the false discoveries (Wilcoxon test adjusted by Benjamini-Hochberg method), then we calculated the global FDR by dividing the average of false discoveries by the total discoveries. The FDR obtained was < 0.05 and is now described in the methods and results section.

19. Page 4 it is described that Ig filtering demonstrated a better separation than the PCA. Was it used only 5 peaks in the Ig analysis?

A total of 38 peaks were used in the Ig analysis. We added the peaks used in the supplementary material, Supplementary Table 2.

20. The evaluations of machine learning is another aspect that warrants more information, careful re-assessments, and/or improved clarity. It is hopeful that there is a real signal in the ML, but the machine learning requires more scrutiny. For example, the "validation" on page 4 isn't a true external validation, the model is just retrained on a different dataset. This suffers from the same issues as mentioned above and thus doesn't provide any additional insight in the validity and generalizability of the initial model. If the method was robust and the model was generalizable, no Ig or retraining would need to be required to get good predictions. Additionally, for this evaluation there is even less information available: how many features were retained after Ig, ...? Please describe better.

Following the reviewer suggestion, we removed the step named validation. All data were used in the nested-repeated cross validation. All feature selected during pre-processing are described in the Supplementary Table 2. Now, the Ig selection is done only in the training set, to avoid information leakage.

In order to prove the validity and generalizability of the model, we prepared 30 new samples classified by the same clinical parameters as high and low risk and applied the experimental and computational methods described in the main text. The text reported below is for reviewer purpose only and it is intended to highlight the robustness of the model:

To address how the model would behave a blind set, we prepared 30 samples (15 high risk and 15 low risk) and applied to the model as a test set. The samples were prepared and the spectra was obtained as described in the methods section. The samples used for training the model (117) were loaded, spectra range were trimmed (2.5 to 15 kDa). The resulting files were transformed (square root method) and smoothed (Savitzky-Golay method), and the baseline correction was done by the TopHat algorithm. Intensities of all files were normalized (total ion current calibration method), and the peaks were detected with a signal-to-noise of 2 and a halfWindowSize of 10. The same procedure was done separately for the blind set samples (30). Next, the peaks were binned (tolerance of 0.003) separately for the high risk and low risk groups of the training set, since the group information of blind test is absent, the peak binning was done in the entire set. Then, the peaks were filtered separately for each subset (training high risk, training low risk and blind set) keeping the ones present only in 80% of the samples. To avoid assigning a peak region to different bins in the subsets, another binning step (tolerance of

0.003) was performed with all samples. Next, the datasets were separated (training set comprising high and low risk samples, and a blind set), the normality was accessed by a Shapiro-Wilk test and a two-tailed Wilcoxon rank sum test with multiple hypotheses testing using Benjamini-Hochberg correction performed only in the training set. The resultant peaks (features) were used for model training.

We trained the RF model with the training set under the optimized parameter of $mtry = 4$ (retrieved accordingly to the main text) and tested if it could classify correctly the new samples. We achieved an accuracy of 90%, a sensitivity of 92.8% and specificity of 87.5% .

21. There is a concern about data leakage as based on the description provided the machine learning was not quite done correctly. For example, the lg is performed on all the data prior to splitting the data into train and test sets for training the machine learning models. This will lead to information the data into train and test sets for training the machine learning models. This will lead to information leakage to the test set, resulting in optimistically biased model performance. Suggested solution: lg needs to be integrated in the model evaluation and the whole pipeline needs to then be properly evaluated on a test set. It could also be that we misinterpreted the description of the methods, and in the absence of the code and the data input this could not be verified. If it was a misinterpretation then clarify and provide the data tables and code.

We performed lg filtering after data split as suggested by the reviewer, so peaks were selected avoiding information leakage between training and test datasets.

22. How does the 10-fold cross-validation samples ordered for evaluation? Were the 10-fold cross-validation with the COVID-19 samples ordered first and the healthy samples ordered second? Explain. In that case for the first few folds the test set will only contain COVID-19 samples and the final few folds will only contain healthy samples. Thus, the evaluation will be completely biased towards one of the two classes. As a suggestion for the authors: randomized stratified splitting should be used.

The samples were split randomly.

23. Please provide hyperparameters of the machine learning models.

The best hyperparameters of each fold is available in the supplementary material.

24. Data and code availability: The SI is unacceptable and not compatible with a good magazine quality. There is no FID data from MALDI-MS spectra. At a minimum they should be the exl, csv or tab-delimited file so it can be efficiently reused. It should need a description of what the SI is in the first place. The data from this study has the potential to become very useful for people throughout the world and should be deposited publicly in a proteomic or metabolomics repository or generalist repositories and can get its own DOI in several such

repositories for the data set. Ideally the data is in both the raw format and open format. If data is not allowed to be made available provide explicit reasons. All new code needs to be accessible, the data tables to make the figs and the scripts to create them. Also have the authors considered creating a systematic web to link data information about the data where this is stored in public repositories where people throughout the world can then compare their data to everyone else's data, including for this study. I envision there will be a lot of interest in this from the wider diagnostic community as the MALDI systems are so widely available. One way may be through the PRIDE repository as they have make such data available and accessible.

We thank the reviewer for the comment. We have made available in the PRIDE repository the LC-MS/MS and MALDI-MS data. These information are open and available to everyone. Moreover, we are appending below the code used to analyze the data. We are working on creating web platform to collect all the information.

I want to iterate that the potential of the tools can be transformative and that I am enthusiastic about the approach but that it will need a lot of points to address and there are hints that are very encouraging but the work needs more refinement. And if after the refinement the results still hold then this work should be published ASAP. I will be happy to rereview in an expedited manner the corrected version.

May 18, 2021

RE: Life Science Alliance Manuscript #LSA-2020-00946-TR

Prof. Giuseppe Palmisano
University of Sao Paulo
Department of Parasitology
Avenida Lineu Prestes 1374
Sao Paulo, Sao Paulo 05508-000

Dear Dr. Palmisano,

Thank you for submitting your revised manuscript entitled "Prognostic accuracy of MALDI-TOF mass spectrometric analysis of plasma in COVID-19". We would be happy to publish your paper in Life Science Alliance pending final revisions necessary to meet our formatting guidelines.

As you will note from the reviewer's comments below, the reviewer is mostly satisfied with the revisions, but has requested an important text edit, acknowledgement of work done in a previous paper, which should be included in the final revision.

Apart from the points listed below, please also attend to the following:

- please consult our manuscript preparation guidelines <https://www.life-science-alliance.org/manuscript-prep> and make sure your manuscript sections are in the correct order
- please add an Author Contributions section to your main manuscript text
- please upload your main and supplementary figures as single files
- please add callouts for Figures 1A-C; 2A,B; 4A-D; 5A,B; S6A-C; S8A,B to your main manuscript text
- please add your main, supplementary figure, and table legends to the main manuscript text after the references section
- we see that you have included the public repository information where the datasets are deposited within different sections of the Methods. We suggest you make a separate 'Data Availability' section in the manuscript text and include the repository information and accession numbers there, as it is usually easier for the readers to obtain this information when it is under a separate section.

A. FINAL FILES:

B. MANUSCRIPT ORGANIZATION AND FORMATTING:

Sincerely,

Shachi Bhatt, Ph.D.
Executive Editor

Reviewer #2 (Comments to the Authors (Required)):

MAJOR COMMENTS

Before this PI can be accepted, I suggest the following Major Correction.

The methodology was optimized from Nachtigall et al (reference 16) that employed nasal swabs to differentiate positive from negative SARS-CoV-2 samples. However, authors did not mention it throughout the text as the starting point by using MALDI-Machine Learning approach. The goal of this PI is different, but it is worth to mention that the effort of Nachtigall (ref 16, Nat Biotechnol.) should be pointed out.

Thus, I suggest to rewrite the last paragraph in "Introduction" page 2. This will not decrease the value of author's work. It is also fair to recognize the methodology studied previously that was optimized in this PI. So, it is missing this recognition in the introduction.

"Although it was previously described the use of MALDI-MS and machine learning analyses in COVID-19 nasal swabs samples,¹⁶ our study shows a new approach for the identification of a plasma proteomic signature obtained from high (hospitalized) versus low (outpatients) risk patients with COVID-19 using an accurate..."

Minor corrections

- Abstract: fingerprinting
-
- Table 1 and 2, p=0.2 and not p=0,2; 11.2 and not 11,2 as well as 6.5 and not 6,5. Page 3,4; Please, correct numbers throughout the text.
- Fig. 5 change 15kDa by "15 kDa) with space. There is always a space between numbers and units. Correct throughout the Text
- Change m/z by "m/z"
- Page 9: change 1x10e6 by "1x10⁶". Correct throughout the text.

May 31, 2021

RE: Life Science Alliance Manuscript #LSA-2020-00946-TRR

Prof. Giuseppe Palmisano
University of Sao Paulo
Department of Parasitology
Avenida Lineu Prestes 1374
Sao Paulo, Sao Paulo 05508-000
Brazil

Dear Dr. Palmisano,

Thank you for submitting your Research Article entitled "Prognostic accuracy of MALDI-TOF mass spectrometric analysis of plasma in COVID-19". It is a pleasure to let you know that your manuscript is now accepted for publication in Life Science Alliance. Congratulations on this interesting work.

DISTRIBUTION OF MATERIALS:

Again, congratulations on a very nice paper. I hope you found the review process to be constructive and are pleased with how the manuscript was handled editorially. We look forward to future exciting submissions from your lab.

Sincerely,

Shachi Bhatt, Ph.D.

Executive Editor

Life Science Alliance

<http://www.lsjournal.org>
